# Short-Term Effects of Forest Therapy on Mood States: A Pilot Study

**DOI:** 10.3390/ijerph18189509

**Published:** 2021-09-09

**Authors:** Francesco Meneguzzo, Lorenzo Albanese, Michele Antonelli, Rita Baraldi, Francesco Riccardo Becheri, Francesco Centritto, Davide Donelli, Franco Finelli, Fabio Firenzuoli, Giovanni Margheritini, Valentina Maggini, Sara Nardini, Marta Regina, Federica Zabini, Luisa Neri

**Affiliations:** 1Institute of Bioeconomy, National Research Council, 10 Via Madonna del Piano, I-50019 Sesto Fiorentino, Italy; lorenzo.albanese@cnr.it; 2Italian Alpine Club, 19 Via E. Petrella, I-20124 Milano, Italy; franco.finelli55@libero.it (F.F.); giomarghe@yahoo.com (G.M.); 3Local Public Health Authority, AUSL- IRCCS, I-42122 Reggio Emilia, Italy; michele.antonelli@ausl.re.it (M.A.); davide.donelli@ausl.re.it (D.D.); 4Institute of Bioeconomy, National Research Council, 101 Via Gobetti, I-40129 Bologna, Italy; rita.baraldi@cnr.it (R.B.); luisa.neri@ibe.cnr.it (L.N.); 5Pian dei Termini Forest Therapy Station, 2311 Via Pratorsi, I-51028 San Marcello Piteglio, Italy; ricerca@terapiaforestale.it; 6“Cesare Alfieri” Political Science School, University of Florence, 60 Via Vittorio Locchi, I-50141 Firenze, Italy; francescocentritto08@gmail.com; 7CERFIT, Careggi University Hospital, I-50134 Florence, Italy; fabio.firenzuoli@unifi.it (F.F.); valentina.maggini@unifi.it (V.M.); 8A.M.I.S.I. Italian Medical Association for the Study of Hypnosis, 28 Via Paisiello, I-20131 Milano, Italy; nardini_s@libero.it; 9Mindfulness Association UK, Edinburgh EH9 1AR, UK; marta@martaregina.com

**Keywords:** anger, anxiety, confusion, depression, forest therapy, mental health, volatile organic compounds

## Abstract

Immersion in forest environments was shown to produce beneficial effects to human health, in particular psychophysical relaxation, leading to its growing recognition as a form of integrative medicine. However, limited evidence exists about the statistical significance of the effects and their association with external and environmental variables and personal characteristics. This experimental study aimed to substantiate the very concept of forest therapy by means of the analysis of the significance of its effects on the mood states of anxiety, depression, anger and confusion. Seven forest therapy sessions were performed in remote areas and a control one in an urban park, with participants allowed to attend only one session, resulting in 162 psychological self-assessment questionnaires administered before and after each session. Meteorological comfort, the concentration of volatile organic compounds in the forest atmosphere and environmental coherence were identified as likely important external and environmental variables. Under certain conditions, forest therapy sessions performed in remote sites were shown to outperform the control session, at least for anxiety, anger and confusion. A quantitative analysis of the association of the outcomes with personal sociodemographic characteristics revealed that only sporting habits and age were significantly associated with the outcomes for certain psychological domains.

## 1. Introduction

The direct exposure to forest environments was attributed a wide range of straight benefits to human health. Such benefits cover primarily the psychological sphere (mental processes, stress, anxiety and emotions), cognitive processes, social life (skills, interactions, behavior and lifestyle) and spiritual wellbeing [1,2,3,4]. On the physiological side, significant effects emerged with regards to the improvement of the cardiovascular functions and the hemodynamic, neuroendocrine, metabolic, immune, inflammatory and oxidative indices [5,6].

As pointed out by a recent comprehensive review, the impact on the psychophysical well-being of humans by immersion in forest environments was by far the most studied and revealed statistically significant and sometimes intense results [7]. However, in the same review, the results of experiences with patients affected by various diseases and addictions were reported, such as improving sleep quality in patients with gastrointestinal cancers [8]. In addition, in-patients with affective and psychotic disorders received significant benefits to their mental health from forest therapy programs, with the type and extent of the improvements related to the characteristics of the given disorder [9].

Individual benefits can turn into benefits for the society as a whole due to cost savings for healthcare systems, insurance and safety, as well as gains in productivity. A recent study showed that the economic value of natural protected areas, assessed on the basis of the mental health of visitors, amounts to about 8% of the global gross domestic product, i.e., around USD 6 trillion [10]. Such an amount, being 100 to 1000 times greater than the budget of the agencies managing the natural protected areas, could justify any scientifically based effort aimed at increasing the efficiency and expanding the functions and ecosystem services of the natural and protected areas towards human mental and physiological health [10].

The preventive and healing effects directly delivered by forest environments have been widely documented in the scientific literature, and, in most cases, concern experiences of free walking or meditation in the absence of appreciable physical exercise [11]. The beneficial effects towards the psychological and physiological aspects of human health are often significant, dose-dependent (e.g., duration and frequency of the experiences and concentration of airborne bioactive substances), related to the structure and tree species composition of the forest and to the season and time of the day [11]. Personal characteristics such as gender, age, psychological traits and physical health contribute to modulating the extent of beneficial effects of the forest immersion. In particular, a recent study performed in Hungary showed that walking in the forest for 2 h, with minimal guidance and aimed at maintaining a slow pace, resting, observation, breathing and touching the wood produced a significant decrease of systolic blood pressure both in late spring and winter [12]. As well, a significant stimulation of NK^bright^ cells and activation of various immune cell subsets was observed in late spring, while a slight activating and a balancing effect regarding TIM-3, an inhibitory immune checkpoint molecule, were observed in winter, thus suggesting positive forest healing effects throughout the year [12].

The forest healing effects can be wide-spectrum, such as in the case of the immune system [13], long-lasting and devoid of undesirable side effects. Just 15–20 min of forest immersion are enough to convey, at least, significant psychological benefits, two hours (even not continuously) allow enhanced wellbeing for a week, irrespective of individual characteristics such as age and presence of chronic diseases [14]. Three or more days of forest immersion can reinforce the immune defense system for a month [15].

An additive effect (constantly growing benefits) deriving from the regular repetition of sessions was observed, which was larger in the case of subjects with depressive tendencies [16]. A recent study, consisting of free forest immersion sessions (four weekly sessions each lasting four hours), suggests the persistence of psychological benefits (indicators of stress, wellbeing and positive emotions) up to a month after the end of the sessions, contrary to virtual immersion experiences [17]. Indeed, virtual experiences showed significant psychological effects but only in the short term [18].

Forest experiences, where professionals assist participants according to structured protocols, often based on psychotherapy, meditation and mindfulness practices [19], showed significant and remarkable therapeutic superiority over both similar activities carried out in non-forested environments and free immersion in forest, for example, in the treatment of depressive symptoms [11,20]. The actual forest therapy foresees not only the individual free immersion in a forest environment, often called forest bathing, but also the guidance by suitable professionals, such as psychologists and psychotherapists, according to protocols aimed at boosting the health outcomes [11,20]. Indeed, the very possibility of issuing green prescriptions in the frame of medical phytotherapy protocols was conditioned on the availability of forest therapy providers [21].

The benefits of forest immersion are mediated by all human senses. Viewing repetitive natural structures (fractals), such as trees, and hearing the typical sounds of forests convey the perception of a relaxing and unchallenging environment, in which it is much easier to adapt and increase one’s own perceptive fluidity [22,23], even in virtual experiences [18]. Touching the wood of the trees induces significant physiological and psychological relaxation [24].

Smell plays a key role through the inhalation of biogenic volatile organic compounds (BVOC) emitted by plants and soil in the forest atmosphere, in particular, certain terpenes endowed with antioxidant and anti-inflammatory activities, as well as beneficial to psychological and cognitive processes [25]. First of all, the concentration and type of such substances in the forest atmosphere depends on the emission rate by plants and thus on the tree species [26] and their vegetative status (thus, the season) [27]. At any forest site and in any season (especially during the warm season), the time of the day plays a very important role due to changing atmospheric conditions (temperature, solar irradiance and surface stability), leading to distinct and regular peaks and troughs in the total BVOC concentration levels [28,29].

The benefits and potential of forest therapy were officially recognized by the United Nations in the framework of the green recovery from the COVID-19 pandemic [30]. The main challenges were identified in informing, educating and persuading healthcare systems and professionals and citizens with regard to green prescriptions, for example, based on those that have been issued for years in Japan and South Korea [5,31,32]. In those countries, as well as in Taiwan and China, green prescriptions have been framed into preventive medicine, as a means to improve physiological health and ease stress levels [33]. Another urgent challenge was identified in the certification of professional providers of services related to forest therapy meant to effectively enable green prescriptions and allow interested people to access a widespread and high quality offer [21,34].

Despite the growing scientific evidence, many countries, especially in the Western world, lag behind in the rigorous identification of sites and trails for forest therapy practices, including preliminary selection and qualification based on actual functionality (observed psychological and physiological benefits) according to structured programs of activities. First, the following consolidated evidence was seldom incorporated: while exposure to most greenspaces allows for improvements in mood and wellbeing, specific and significant preventive effects with regard to the risk of depression were attributed only to natural environments, in particular forested ones, remote from urban areas [35,36]. This is particularly important because depression is a pandemic illness, especially in the developed world [35], with immense human, social and economic costs [36]. Thus, natural remedies such as forest therapy performed in suitable locations can lead to substantial savings for healthcare and welfare systems and gains in terms of productivity [34].

Second, especially in the Western world, forest therapy experiences in remote forest areas involving large numbers of participants, according to structured and repeatable programs and including control sessions in urban settings, were seldom performed, let alone the search for the main determinants of health outcomes that would allow the generalization of qualification criteria. Despite recent improvements in the methodological rigor, the recommendations issued in a comprehensive review published in 2017 are still relevant [37].

The main motivation of this study was to quantitatively assess the effectiveness of a few forest therapy sessions performed in remote areas on the improvement of certain mood states, i.e., anxiety, depression, anger and confusion, compared with a control session, thus substantiating the very concept of forest therapy. Another aim was to contribute to the preliminary identification of the main determinants of the outcomes of forest therapy sessions, taking advantage of the different environments involved, the relatively large number of participants and the representativeness in terms of sociodemographic characteristics.

The main hypotheses of this study were: (1) forest therapy carried out in remote forest areas is more effective than in urban parks; (2) forest therapy (professionally guided) is more effective than forest bathing (free); (3) different environmental conditions, either structural or variable, and personal characteristics, affect the outcome of forest therapy sessions.

## 2. Materials and Methods

### 2.1. Forest Therapy Sessions: Basic Features and Measures

The Italian Alpine Club and the Institute of Bioeconomy of the National Research Council arranged seven forest therapy sessions from August to October 2020. All sessions, named FTS1 to FTS7, were performed in Italy at remote forest areas (at least 20 km from the nearest town with more than 10,000 inhabitants or trafficked road). All sessions had a duration of 3.5 ± 0.5 h, along paths with total lengths not exceeding 4 km and uphill gradients smaller than 150 m.

A further control session, named FTSC, was performed in June 2021, in an urban park with an area of about 4 hectares, located downtown in the city of Florence, Italy (about 400,000 inhabitants) and surrounded by buildings and trafficked roads.

Figure 1 shows the location of the forest therapy sites in central and northern Italy.

During each session, the concentration of total volatile organic compounds (TVOCs) was continuously collected by means of a portable (0.72 kg) photoionization detector (PID, model Tiger VOC detector, Ion Science Ltd., Fowlmere, Royston, UK), with detection limits from 1 μg/m^3^ to 20,000 μg/m^3^. The PID was equipped with a pump, sucking in ambient air at a rate of 220 mL/min, and an ultraviolet, 10.6 eV lamp, allowing the ionization of organic substances present in the sampled air, including all the monoterpenes (MTs) found in biogenic VOCs (BVOCs). The resulting electric current flowing between two electrodes was measured, amplified and transformed into a concentration level of the ionized substance or group of substances. TVOC concentration could be considered representative for BVOC concentration, although clearly higher due to the presence of VOCs other than MTs, at least at forest sites far from anthropogenic pollution sources where the change in TVOC concentration is mainly driven by plants’ emissions, as explained in a previous study [28]. Finally, the average temperature during each session was assessed based on data collected at nearby weather stations.

The first session (FTS1), which can be classifiable as forest bathing, was performed without professional guidance, suggesting to participants to relax in the woods during two stops lasting between 20 and 30 min each, and providing scientific information about the surrounding environment (botany and geology) and the sessions meaning for one’s own health. All the other sessions were performed with professional guidance, according to the method of conducting illustrated in Section 2.2. However, the control session FTSC, which was performed in an urban park downtown the city of Florence, Italy, although including professional guidance, was carried out according to a method of conducting that was different and substantially simplified in comparison with the method illustrated in Section 2.2, in particular based on the immersion in nature with mindful use of the senses of sight, hearing, touch and smell.

Volunteer participants were enrolled and insured by local sections of the Italian Alpine Club following information spread by means of standard channels (newsletters, social media, etc.). No selection was applied except for age (>18 years) and the requirement of being healthy. Participants were allowed to attend only one session, to prevent any contribution from the possible persistence of the effects. Overall, 207 participants were recruited, returning 162 valid self-assessment questionnaires, the nature of which is described below in Section 2.1. Table 1 shows the basic information about forest therapy sessions. 

Figure 2a–f shows representative pictures of forest therapy sites.

Limited to participants that returned valid questionnaires and relevant demographic data, there were more female (84) than male (50) participants, the age group 41–60 was the most represented for both genders (almost half of total) and the place of residence of more than 59% of participants was outside towns with more than 40,000 inhabitants. Figure 3a,b shows the basic demographic features of participants to forest therapy sessions.

Ryan–Joiner test for normality was performed for height and weight variables for the entire study population, resulting in a non-normal distribution (height: *p* = 0.044; weight: *p* < 0.01). Moreover, chi-square test matrix for homogeneity among categorical variables was performed for the entire study population for sex, study degree, job, smoking and residence variables, resulting in homogeneity only for sex variable (*p* = 0.182). Table 2 shows the data about age, height and weight for each forest therapy session.

Participants reached the forest therapy site by car, and the informed consent was collected on site and in written form from every participant before each session. Overall, at least half an hour, and up to one hour, elapsed before participants started the forest therapy session, during which every participant was presented also with a reduced Profile of Mood States (POMS) questionnaire and listened to a short speech. Such time could have helped relieving transport-related stress. The POMS questionnaire was completely anonymous and included information about gender, age, place of residence, job and habits such as smoking and sports. The POMS questionnaire consisted of 34 questions about moods perceived at that precise moment, recommended for the analysis of psychological outcomes of forest therapy sessions [38] and derived from previous studies on sport psychology [39]. The same questionnaire was recently used in an investigation of stress reduction of healthcare staff after a short break in nature [40]. However, since no validation study could be found regarding the specific subset of POMS items, it was decided to limit the analysis to the psychological domains for which all the items were also included in the originally validated questionnaire [39]. Thus, the considered domains were the mood states of anxiety (also called tension in the original version), depression, anger and confusion according to the recommended classification [39]. The POMS questionnaire was filled immediately before and after the session. For each questionnaire, single answers to any of the items were grouped to provide the above-mentioned synthetic indices. Only completely filled questionnaires were considered for further analyses.

### 2.2. The Method of Conducting the Forest Therapy Sessions

Except for session FTS1, and for session FTSC that was carried out according to a different and simplified method of conducting referred to in Section 2.1, a professional psychologist or psychotherapist took part in each of the other forest therapy sessions, according to a program named Forestfulness^®^, partly based on mindfulness practices, in particular on the Mindfulness-Based Stress Reduction (MBSR) program developed since 1982 [19]. Mindfulness-based meditation is a way to pay attention to the present moment by observing and accepting whatever experience is occurring.

Mindfulness was shown to reduce pain in a wide range of clinical and non-clinical problems [41], improve immune functions [42], improve mental health in breast cancer patients [43], create a higher subjective wellbeing and attenuate emotive reactivity [44], reduce relapses in depression and blood pressure [45], reduce anxiety and improve mood state [46]. Mindfulness practiced in forest environments showed significant and remarkable therapeutic superiority over similar activities carried out in non-forested environments, for example, in the treatment of depressive symptoms [20].

In the forest environment, mindfulness can be practiced by means of walking mind-fully, using breathing as the center of attention, observing the surrounding environment with acceptance and creating a deep connection with nature using the five senses. Walking meditation is one of MBSR’s main practices, while walking in forests can lower blood pressure and reduce the concentration level of cortisol (also named the stress hormone), resulting in relaxation [47,48]. Walking meditation is performed by bringing the attention to the physical sensations coming from breathing and walking. Although this exercise could produce good results also indoor or in urban environments, higher benefits were observed in the woods, including improvement in mood state [49].

Conscious and focused use of all human senses—sight, hearing, smell, touch and taste—is another milestone of the MBSR program as a means to increase the awareness of the experience and amplify the effects of forest immersion. Through the perception of the surroundings, senses provide orientation, rooting into the present an interpretation of our experiences [50,51]. Indeed, benefits for health from the exposure to forest environments are effectively conveyed by the mediation of all human senses [33,52,53], which further motivates the conscious focusing on human senses during forest therapy sessions.

Moreover, in the conduction of the forest therapy sessions, a method of communication was adopted, which alternated metaphorical speeches with literal verbalizations in order to stimulate increased emotional involvement in the participants [54]. In the past, the use of metaphor was considered more an art than a scientific instrument [55], although one with therapeutic value [56]. However, more recently, this classification has been questioned, with interesting practical implications. For example, it was established that the educational metaphors are useful for simplifying difficult concepts during class lessons [57]; in the clinical field, metaphors help in understanding delicate health conditions [58] and can be useful for developing scientific theories [59]. In everyday life, metaphorical language is a tool that is able to promote and guide thought processes and to activate cognitive maps for problem solving [60]. From a therapeutic point of view, metaphorical discourses represent a cornerstone of Ericksonian and Neo-Ericksonian Hypnotic Psychotherapy; for example, a recent study demonstrated the effectiveness of Mindful Hypnotherapy to reduce stress and create mindfulness [61].

In the context of forest therapy sessions, the attention of the participants to the present moment was stimulated in order to promote a more effective immersion, in addition through the conscious use of the five senses, the breath and the movement, looking for the deep connection with the surrounding environment. Both mindfulness practices and metaphorical discourses contributed to this purpose, as they could anticipate or deepen the proposed experience at an imaginative and evocative level and prepare neural maps for what will be physically practiced. Some neuroimaging studies showed that metaphors activated cortical areas corresponding to the suggested action; for example, the expression “she grabbed the idea” could activate the motor cortex [62].

During the structured forest therapy sessions (FTS2 to FTS7 listed in Table 1), in order to stimulate the sensory activities, the participants were invited to “savor the experience”, to “touch with the eyes the surrounding environment” and to “listen to their emotions”, to name a few. The words chosen in these verbalizations aimed at several purposes:Generating a sensation of coherence between the internal dimension (feeling) and the external dimension (acting calmly). The sense of coherence (internal–external), in accordance with temperament and emotional intelligence, seems predictive of the individual’s ability to cope with stressful situations [63];Facilitating processes of introspection and decoding of the feelings and emotions of the participants to feed a condition of conscious presence;Promoting processes of identification with the elements of nature, which are in dynamic balance and can offer more effective learning and working models [64];Stimulating the transformative activity of any dysfunctional aspects detected [65].

The method of conducting the forest therapy sessions was structured in six main steps, briefly:1.Right attitude and entrance into the woods: Leave behind challenging thoughts, and smile as if meeting with an old friend;2.Wearing the most beautiful dress: Embrace the silence, leave behind planning and commitments, gain space for the emergence of one’s own best part;3.Walking mindfully: Involves bringing attention to any sensation coming from the feet in order to keep the wandering mind anchored;4.Breathing in the woods: By observing the breathing, it is possible to tune in with the environment and to be aware of the state of mind and body as well; consequently, let it all go, such as tensions, thoughts and feelings;5.Using the senses: Intentionally looking at the surrounding environment or a natural object that is found to be relaxing, recognizing wood fragrances during breathing, concentrating on typical forest soundscape (birds, wind through the trees, crunch of branches and leaves underfoot, etc.), touching wood and trees and, when possible, tasting products of the undergrowth;6.Meditating with the trees: Choosing a specific tree as the most faithful mirror of oneself and approaching it with the right attitude and intentions, towards the identification with the tree and its qualities such as resilience, vitality and projection towards the sky and the light.

### 2.3. Data Analysis

Data reported on the reduced POMS questionnaires described in Section 2.1 were transferred to Microsoft Excel worksheets. Psychological indices were obtained from the questionnaires’ data by adding the values of the corresponding items according to the recommended classification [39]. In addition, the TVOC data collected from the PID were transferred from the proprietary software of the instrument to Microsoft Excel, and average and peak concentration levels along the forest therapy paths were computed.

Pre-post average percentage differences of the outcomes for each considered mood state were computed for each session based on data stored in Microsoft Excel worksheets. Then, Minitab 19 was used for further statistical analyses. One-way ANOVA was preferred since the samples were small in number and with quasi-normal distributions. For non-normal samples, the Mann–Whitney test was used. The Ryan–Joiner test was used to test for normality of continuous variables, while the chi-square test was used to test for homogeneity among categorical variables. An ANOVA mixed-effects model was used to test for associations between pre and post changes of each POMS domain and sociodemographic features.

## 3. Results

### 3.1. Overall Psychological Outcomes

In order to ensure that the comparison among different sessions would not be affected by statistically significant differences in initial mood states (i.e., baseline), a one-way ANOVA test was performed on the pre-intervention POMS questionnaires [39] across all forest therapy sessions including the control session (FTSC). The results showed no statistically significant differences among groups in terms of pre-intervention scores for all mood states: anxiety (DF = 7; DF error = 154; F = 1.59; *p* = 0.141), depression (DF = 7; DF error = 154; F =1.66; *p* = 0.124), anger (DF = 7; DF error = 154; F = 0.64; *p* = 0.725) and confusion (DF = 7; DF error = 154; F = 1.71; *p* = 0.111). Table 3 shows the relevant statistics.

The differences in scores for any mood states before and after forest therapy sessions were tested; in particular, data were aggregated for intervention groups (FTS1 to FTS7), while FTSC was tested separately. A one-way ANOVA was performed to test for pre–post differences in the aggregated intervention groups (FTS1 to FTS7) and in the control group (FTSC), resulting in a statistically significant reduction in each POMS domain score for the aggregated intervention groups, while the control group resulted non-significant for each POMS domain. Table 4 shows the results, along with the respective F and *p* values. The control group had a limited numerosity; thus, the observed trends toward a reduction in any of the mood states might become statistically significant with a larger sample, especially for anxiety (*p* = 0.191).

### 3.2. Specific Effects of Remote Forest Areas

The possible specific effects due to the remoteness of forest areas and the conditions under which such effects could arise were investigated by means of the comparison of outcomes (pre–post differences, i.e., change-from-baseline) of each intervention group (FTS1 to FTS7) with the corresponding outcomes from the control group in the urban park (FTSC).

First, Grubb’s test for outliers was performed on the changes-from-baseline for all mood states derived from the answers to the POMS questionnaires [39] in order to exclude potential outliers from the analysis (*p* < 0.05). For anxiety, only one outlier was removed from session FTS7. For depression, one outlier was removed from each of the following sessions: FTS1, FTS2 and FTS6. For anger, one outlier was removed from each of the following sessions: FTS1, FTS2, FTS5, FTS6, FTS7 and FTSC. For confusion, only one outlier was removed from session FTS1.

Next, a Ryan–Joiner test for normality was performed for the change-from-baseline of each mood state derived from every forest therapy session. Table 5 shows the results of the test, where only non-normal distributions are highlighted (*p* < 0.05).

A one-way ANOVA test was applied for comparing normal samples, while a Mann–Whitney test was applied when the comparison involved at least one non-normal sample. Table 6 highlights the cases in which a forest therapy session performed in a remote forest area resulted in improvements of mood states significantly higher than in the FTSC (*p* < 0.05) or close to significantly (0.05 ≤ *p* < 0.10). The latter were added because the limited number of participants to certain forest therapy sessions might have hindered the achievement of statistical significance.

Sessions FTS1, FTS2 and FTS5 showed no specific additional benefit compared to the control session FTSC. Session FTS3 showed an improvement in anger significantly higher than FTSC and comparative improvements in depression and confusion close to significance. Session FTS4 again showed an improvement in anger significantly higher than FTSC and a comparative improvement in confusion close to significance. Session FTS6 showed improvements in anxiety and confusion significantly higher than FTSC and a comparative improvement in anger close to significance. Finally, session FTS7 showed an improvement in anxiety significantly higher than FTSC and a comparative improvement in anger close to significance.

### 3.3. Association of Pshycological Outcomes with Sociodemographic Traits

A mixed model ANOVA was applied to check for any statistically significant association among the collected sociodemographic traits of the participants in the forest therapy sessions, excluding the control session FTSC, and the respective changes-from-baseline for any mood state. The variance estimation method was the restricted maximum likelihood and the Kenward–Roger method was used for testing fixed effects. In total, 128 questionnaires were considered, reporting complete sets of sociodemographic information. Table 7 shows the considered factors and their respective values.

The results showed a statistically significant association of the change in anxiety with sport activity (*p* = 0.035), in particular, the absence thereof (*p* = 0.020) with a negative coefficient, i.e., anxiety decreased significantly for people who never played sports. Changes in depression were significantly associated with age (*p* = 0.031), with a positive coefficient, as well as with sport (0.038), in particular, the absence thereof (*p* = 0.041), with a negative coefficient. Changes in anger were significantly associated with sports (*p* = 0.011), in particular, the absence thereof (*p* = 0.002), with a negative coefficient. Changes in confusion were not significantly associated with any considered factors, although the association with age was close to significance (*p* = 0.058) with a positive coefficient.

## 4. Discussion

### 4.1. Analysis of Presented Results

Based on the results shown in Table 4, the outcomes (i.e., change-from-baseline) from the intervention groups altogether (forest therapy sessions FTS1 to FTS7, totaling 150 valid questionnaires) were statistically significant for any of the considered psychological domains (anxiety, depression, anger and confusion). Thus, this study supports the hypothesis that forest therapy sessions performed in remote forest areas are effective in improving the considered psychological domains.

Conversely, none of the outcomes from the control group (session FTSC) were significant, suggesting that remote forests, where sessions FTS1 to FTS7 were performed, could bring specific benefits compared to urban forests. However, a main limitation of this study was the small numerosity of the control group (12 valid questionnaires), which might have hampered the significance of the respective outcomes.

In order to try to overcome the above-mentioned limitation and provide further insight into the factors affecting the outcomes of forest therapy sessions, the outcomes from each of the intervention groups were separately compared with the corresponding outcomes from the control group. As presented in Section 3.2 and shown in Table 6, few sessions performed in remote forest areas resulted in specific and significant benefits for some psychological domains, compared with the control session performed in an urban park. In particular, both FTS6 and FTS7 significantly outperformed FTSC for anxiety and close to significantly for anger, while FTS3 and FTS4 significantly outperformed FTSC for anger and close to significantly for confusion. FTS3 also outperformed FTSC for depression, but only close to significantly. FTS6 showed the best results, also significantly outperforming FTSC for confusion. Sessions FTS1, FTS2 and FTS5 did not significantly (or close to significantly) outperform FTSC. Overall, these results suggest that performing forest therapy sessions at remote sites could help to enhance its effects compared to urban settings, but only under certain circumstances.

Excluding FTS1, which was conducted non-professionally, and FTS5, which was affected by a small numerosity and relatively harsh weather conditions, what peculiar characteristics could underlie the additional and significant benefits deriving from certain forest therapy sessions performed in remote sites?

Based on Table 1, FTS6, which was the highest performing session, enjoyed the highest average concentration of TVOCs and was the only one performed in a Mediterranean forest, without water bodies or water streams. On the other hand, FTS3 enjoyed the second highest average concentration of TVOCs (and the highest among peak concentrations) and, similar to FTS4 (performed on the same site as FTS3), included fascinating water streams and small waterfalls. Notably, FTS4 enjoyed very good weather comfort, similar to FTS3, but substantially lower average and peak concentration of TVOCs. FTS7 suffered from less comfortable weather compared to all the other sessions, along with lower average concentration of TVOC, but included water streams and a fascinating natural pond. Finally, FTS2 enjoyed very comfortable weather, concentration levels of TVOCs only slightly lower than FTS4 and higher than FTS7 and did not include water streams or water bodies.

Based on the available data, essentially on the performances of FTS6 compared with all the other sessions and of FTS3 compared to FTS4, the hypothesis could be advanced that the concentration of TVOCs is an effective factor affecting the performance of professionally conducted forest therapy sessions in remote forests. This can be regarded as an original experimental result of this study.

However, the matter is further complicated by partially contrasting results about the systemic absorption of MTs, which represent most of the BVOCs emitted by plants, which, in turn, are likely to contribute substantially to the observed concentration of TVOCs. To the best knowledge of the authors, only two studies dealt with this important topic. In 2015, Japanese scholars found that the serum concentration of α-pinene, but not of other monoterpenes, increased by several folds after walking in a conifer forest for one hour [66]. In 2021, Spanish scholars found that the blood concentration of α-pinene, but not of other monoterpenes, increased by a much slighter extent and only in subjects starting from the lowest baselines levels after walking in an evergreen broadleaf forest (holm oak as the most representative species) for two hours [67].

Although very important and innovative, both studies [66,67] were affected at least by small sample sizes, by the unknown individual levels of metabolic transformation rates and by the limited exposure time. Moreover, the health effects produced by MTs could follow pathways other than by systemic absorption, such as direct effects on the upper respiratory tract and on the central nervous system through brain absorption [25]. However, due to the great importance of α-pinene as an anti-inflammatory, analgesic, antioxidant, anxiolytic, antidepressant and sedative, as well as an antiproliferative agent [25], the results obtained in this study, although quite limited, could point to some impact of BVOC concentration on the psychological outcomes after forest therapy sessions lasting about 3.5 h. Clearly, the detailed analysis of the relative and absolute concentration of the different monoterpenes available in TVOCs during further forest therapy sessions involving large numbers of participants will help clarify the possible impact of different BVOCs on short-term psychological outcomes.

Despite lower levels of weather comfort and concentration of TVOCs, FTS7 generated few significant results compared with the control session FTSC. Conversely, a very good weather comfort and concentration levels of TVOCs substantially higher than in FTS7 and only slightly lower than in FTS4 did not allow session FTS2 to achieve any significant result in comparison to FTSC, underperforming in comparison to all the other sessions. The hypothesis could be advanced that water streams and water bodies play an important role, as already arisen in previous studies, for example, in a controlled experiment using virtual reality [68]. However, considering that FTS6 lacked water bodies or water streams, the matter could be more complicated.

It is further hypothesized that the presence of water streams or water bodies is very important, but only if appropriate to the specific environment. In fact, FTS6 outperformed all the other sessions even in the absence of those elements, but one could not expect their presence in a typically dry Mediterranean forest. Conversely, FTS2 underperformed all the other sessions, and one would expect to find water streams in a mountainous environment dominated by Scots pine. It remains unknown whether substantially higher concentration levels of TVOCs at the site of FTS2 could compensate for a possibly less efficient environment: ongoing research at that site might help clarify the issue. Indeed, FTS2 was performed in late summer, while, for Scots pine forests, it was found that young sprouts in late spring and early summer emit far more BVOCs than mature needles in late summer [27].

Overall, the coherence among the main natural elements, sometimes called environmental coherence [69], could play a distinct role in boosting the performance of forest therapy sessions. This coherence could also help explain why FTS7, carried out in a very humid mountain environment, performed quite well despite low concentration levels of TVOC and limited weather comfort. Moreover, it should be noted that FTS3 and FTS4, despite the difference in the concentration of TVOCs, produced very similar results concerning the psychological domains, i.e., significantly better performances compared with the control session for anger and close to significance for confusion (as well as for depression, but only in FTS3). The specific and coherent environment characterizing FTS3 and FTS4 could have been an effective driver for improvements at least in anger and, likely, in confusion. The possible role of environmental coherence is another original experimental result of this study, although it requires further research in order to elucidate its nature and importance.

Session FTS1, which produced no significant outcome compared to FTSC, was performed in the same site as FTS6, although with a remarkably lower level of weather comfort (temperature of 30 °C compared to 15 °C in FTS6) and according to a non-professional method. FTS1 enjoyed the highest average and peak concentration of TVOCs across all forest therapy sessions. Although the comparison between FTS1 and FTS6 could lead to hypothesize an important role for the method of conducting forest therapy sessions, with professional guidance more effective in producing significant outcomes, the large difference in weather comfort jeopardizes this hypothesis. Further experiments will be needed in order to gain insight into this matter.

Based on Table 6, among the considered psychological domains, depression was the least affected by the remoteness of the forest site, at least in comparison to anxiety, anger and confusion. Possibly, items used to build the depression index would need longer forest therapy sessions or regular repetitions for their effective marginal reduction in comparison to sessions performed in urban settings.

The reasons why anxiety was significantly only reduced in FTS6 and FTS7 in comparison to the control are hard to assess. Possibly, they have to do with the forest environments characterizing FTS6 and FTS7 (distanced, mostly brightly colored trees and relatively low beech trees), which could be regarded as more relaxing, or less aggressive, than in FTS3 and FTS4 (dense, dark-colored spruce trees), but further ongoing field trials will hopefully clarify this issue.

Overall, anger appears as the most affected psychological domain, having been significantly reduced in comparison to the control in FTS3 and FTS4 and close to significantly in FTS6 and FTS7. Possibly, items used to build the anger index were representative of acute conditions and could have been more easily reduced by means of forest therapy sessions performed at remote locations. Finally, confusion was significantly reduced compared to the control in FTS6 and close to significantly in FTS3 and FTS4. Beyond environmental coherence, the concentration level of TVOCs, which was substantially lower in FTS7 compared to FTS3, FTS4 and FTS6, could have contributed to the result.

Concerning the association of the psychological outcomes from intervention sessions (FTS1 to FTS7) with the sociodemographic traits presented in Section 3.3, sporting habits showed the highest degree of association, in particular, the absence of sports. In fact, the latter was significantly and negatively associated with the size of reduction in anxiety, depression and anger. The significant association with the absence of sport could mitigate the missing information on duration and intensity, according to the definition of sports presented in Table 7. An important question arises as to whether such association means that people never practicing sport receive substantial benefits from simply walking, irrespective of the surrounding environment and method of conducting physical activity, or the absence of sport is related to a reduced contact with nature and the immersion in forest environments is the leading factor, irrespective of physical exercise. Further field experiments will help resolve this issue.

Age was the other trait showing some association in particular with depression (significant) and confusion (very close to significant). The benefits for depression and confusion increased with age, which could be ascribed either to increasing trait levels of such indices with age or to a greater sensitivity of older people to forest immersion. Again, further research will help clarify this issue.

It is interesting to note that no other sociodemographic characteristics were significantly associated with the results of this study: neither biometric features such as height and weight, nor sex, job, education, residence and smoking habits. This evidence could mean that forest therapy can be successfully applied to most people, irrespective of their conditions.

### 4.2. International Experiences

A recent systematic review on the effects of nature walks on states of anxiety and depression evidenced clear and significant effects on anxiety but somewhat inconclusive effects on depression [70], in agreement with the results of this study. However, most of previous studies dealing with the short-term effects of forest therapy sessions, in particular, using POMS questionnaires, appear to lack a proper consideration of the nature of the statistical distributions of the changes-from-baseline. Thus, comparisons with the results of this study are difficult to perform.

Research was performed in 2018 based on a 3-day structured forest therapy session in a semitropical forest in Taiwan, mainly aimed at checking the effects on creativity and involving 21 participants (8 males, 13 females) across all age groups (52.0 ± 12.54 years, from 25 to 70 years old) [71]. The analysis of POMS questionnaires returned only the percentage changes from baseline, along with the Cohen’s d index for the assessment of the size of the effect [72]. The following changes were presented: anxiety, −49%; depression, −42%; anger, −38%; confusion, −33%. Based on the above-written, comparisons with this study are worthless, let alone the fact that sessions considered in this study had much shorter duration (around 3.5 h).

In a recent study performed in Italy over healthcare staff during the COVID-19 pandemic, 77 participants filled the same reduced POMS questionnaires used in this study, before and after short breaks (20–30 min) in nature (green areas nearby hospital facilities) [40]. The results showed improvements in all the psychological indices, which were greater for staff directly involved in COVID-19 treatment areas, who were likely more stressed. Throughout all participants, anxiety, depression and anger reduced by 60–66%, 53–56% and 47 to 84%, respectively, while confusion was reduced by 35–53%. Again, comparisons with this study are hard to perform.

In a study performed in Taiwan, 128 middle-aged to elderly subjects were recruited (43 males and 85 females, age range 45 to 86 years, on average 60.0 ± 7.44 years) to attend structured forest therapy sessions focused on the activation of the senses [73]. Each session included around 10 participants during 3 h, such duration being comparable to sessions performed in this study, under conditions of high meteorological comfort. The analysis of reduced POMS questionnaires returned the following changes: anxiety, −59%; depression, −50%; anger, −55%; confusion, −44%. 

A study was performed in Poland on young adults (21 participants, of which 12 were males and 9 were females, with an age range between 21 and 29 years and an average age 23.86 ± 2.67 years) according to a forest therapy program structured over two days (indoor and in forest) [74]. The analysis of reduced POMS questionnaires returned the following changes: anxiety, −56%; depression, −54%; anger, −43%; confusion, −36%.

Another study was performed again in Poland on young adults based on structured 30-min walks [75]. The analysis of the reduced POMS questionnaires, referring to a conifer forest, returned the following changes: anxiety, −53%; depression, −35%; anger, −34%; confusion, −30%.

In a study performed in South Korea, 38 university students with a mean age of around 22 years, with 24 males and 14 females, were assigned in equal numbers to a forest therapy group or a control group [76]. The experimental group performed eight, one-hour-long sessions once a week according to a structured program but without active guidance. Reduced POMS questionnaires were presented to the participants at the beginning of the first session and the end of the last session. Except for anger, all other indices improved substantially: anxiety (−34%), depression (−42%) and confusion (−31%).

In the opinion of the authors, it would be useful to re-examine some of the above-cited studies based on appropriate and standardized statistical methods in order to build a really useful international database concerning the performance of forest therapy and the determinants of the respective outcomes.

### 4.3. Limitations and Preliminary Technical Guidelines

A few important limitations affected this study. All the forest therapy sessions lasted 3.5 ± 0.5 h, thus, the dose effect could not be investigated. All professionally conducted forest therapy sessions in remote forest areas followed the same program, thus the effect of different programs could not be investigated. In this respect, a recent Korean study performed with elderly individuals showed that forest therapy programs focused on Qigong or active walking produced specific neuropsychological and electrophysiological benefits, both beneficial for preventing dementia and relieving related health problems [77]. Moreover, the single session performed without professional guidance (FTS1) was affected by a low level of weather comfort, which could have played an important role in the respective insignificant results.

Participants in the forest therapy sessions discussed in this study were allowed to attend only one session, thus the frequency effect could not be investigated. Neither the psychological trait conditions nor the follow-up conditions of the participants were collected; thus, the possible persistence of effects could not be investigated either. Another limitation was that participants were not asked about their personal frequentation of forest or green areas, even if the collected information about personal sporting habits might give some clues to this effect. The latter limitation could have introduced a bias in terms of persistence of effects for some sessions. Moreover, further information on the duration and intensity of sport, understood here as a session of physical activity, would be useful to refine the analysis.

The administration of the POMS questionnaires to the participants allowed collecting instantaneously perceived mood states, not corroborated by any measure on physiological parameters. Only admittedly healthy participants were recruited, thus the effects on individuals affected by psychological or physiological disorders could not be assessed. Moreover, no health screening of participants was performed before the forest therapy sessions, thus it cannot be excluded that some participants were affected at least by acute physiological, psychological or psychiatric illnesses, or by alcohol and drug abuse.

Although representative enough of different outdoor forest environments in central and northern Italy, the study sites could not be considered representative of all the possible environments; for example, chestnut forests, widespread in mid-mountain areas in the Italian Apennines, were not investigated, along with coastal Mediterranean forests. Moreover, elements other than forest species and few other natural features (e.g., water streams or ponds) were not considered, such as artificial elements (fences, tables and chairs, mountain retreats and their architecture, etc.). Although large enough in comparison with other experiences reported in the literature, the investigated sample (sites, sessions and number of participants) could not allow the disentangling of all the possible factors affecting the considered psychological outcomes, as well as adequately representing all the relevant age groups, genders and other sociodemographic characteristics. In particular, the control session FTSC suffered from a relatively scarce number of participants. Finally, only TVOCs concentration was measured, while individual monoterpenes are known to potentially produce different and specialized health effects [25], which could be related to the outcomes of forest therapy sessions.

It is planned to overcome most of the above-listed limitations in the ongoing research across many other sites by using many more participants and including larger control groups in urban settings across different seasons and performing more comprehensive measures such as sociodemographic, psychological (trait and state) and physiological measures, as well as measuring the relative and absolute concentrations of individual BVOCs and MTs. In addition to extending the performance database, the continuation of the study aims at more rigorously identifying the factors affecting the observed healing effects and further linking the medical and forestry research, as recommended in a recent study [11].

Another limitation concerns the considered health domains, which were focused only on mood states. Within the psychological domain, for example, cognitive processes were shown to receive distinct benefits from forest therapy [25,71] and would be worth being investigated, such as currently occurring in the frame of a large European project aimed at fostering knowledge about the relationship between Information and Communication Technologies and Public Spaces [78].

Based on the results obtained in this study and within its limitations, the following preliminary guidelines concerning the short-term amelioration of psychological indices can be advanced. Meteorological comfort, environmental coherence and concentration levels of TVOCs appear quite important in the determination of the outcomes, especially contributing to specific additional benefits brought by forest therapy sessions performed in remote forest areas. Contrary to anxiety, anger and confusion, the depression domain seems not to receive specific benefits from the remoteness of the forest site. Older people could benefit more from forest therapy sessions, especially for depression and confusion. People abstaining from sporting activities, and likely from regular contact with nature, could benefit more from forest therapy sessions, especially for anxiety, depression and anger. These preliminary guidelines could help to optimize forest therapy sessions, as well as to plan suitable sessions targeted to specific groups of participants, for example, those belonging to a given age group or following certain habits, or aimed at improving specific psychological indexes.

### 4.4. Policy Guidelines

The mobilization of financial resources and the introduction of specific legislative elements aimed at promoting, standardizing and regulating forest therapy programs were deemed essential to the effective management of the forest ecosystems towards human health [5]. More specifically, based on the evidence shown in this study, further fundamental challenges consist in the information, education and motivation of health facilities and professionals, as well as of citizens, towards green prescriptions, which have been issued for years in countries such as Japan and South Korea [31].

Further important tasks concern the certification of sites based on structural and environmental characteristics and direct functionality (observation of health effects), as well as the certification of professional providers of services connected to forest therapy [34]. In the Western world, the Parks Prescription program introduced by the British Columbia Parks Foundation in Canada aimed at the prescription of short and slow walks in protected forest areas as a “superfood of exercise” could serve as a useful model [79].

## 5. Conclusions

The results achieved in this study confirm, reinforce and expand the knowledge about the healing effects of forests by means of structured forest therapy sessions. The short-term reductions in the levels of anxiety, depression, anger and confusion were all significant for forest therapy sessions performed in remote areas and taken altogether, contrary to a control forest therapy session performed in an urban park.

Specific effects of remoteness, in comparison to the control session, arose under certain conditions with anxiety, anger and confusion but not for depression. Some characteristics of the site appeared to affect the outcomes, especially the level of meteorological comfort, the concentration of TVOCs and the environmental coherence, as well as personal characteristics of the participants, such as sporting habits and age.

Despite several limitations, a few preliminary technical guidelines were issued, which could help to identify, design and optimize forest therapy stations and sessions also targeted to specific groups of participants or improving specific psychological conditions.

## Figures and Tables

**Figure 1 ijerph-18-09509-f001:**
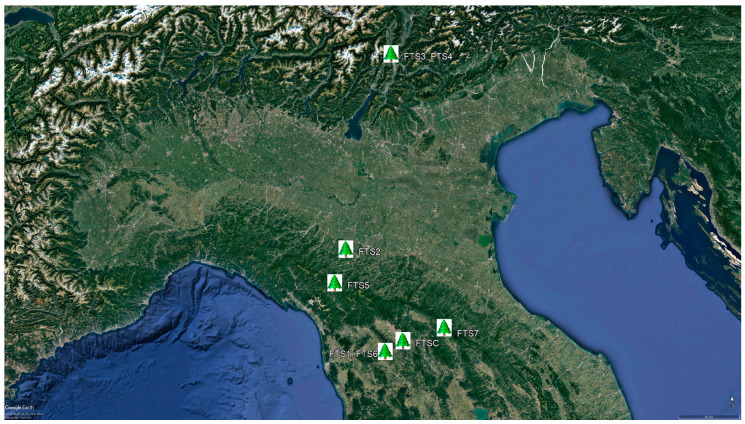
Locations of forest therapy sessions (FTS1 to FTS7 and FTSC) in central and northern Italy.

**Figure 2 ijerph-18-09509-f002:**
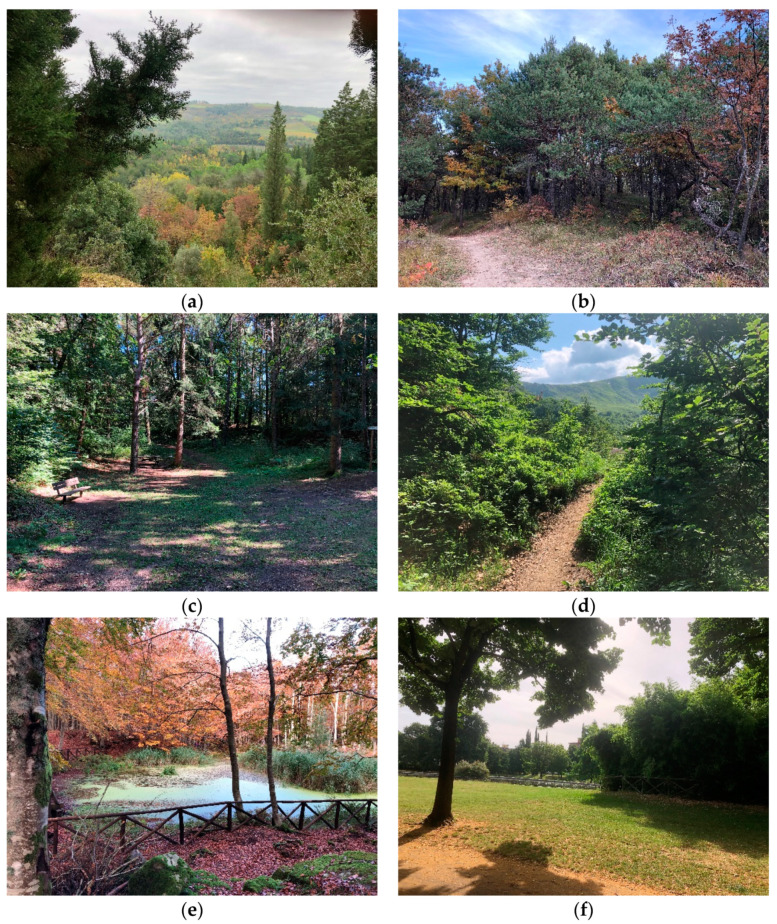
Representative pictures of forest therapy sites: (**a**) FTS1 and FTS6; (**b**) FTS2; (**c**) FTS3 and FTS4; (**d**) FTS5; (**e**) FTS7; (**f**) FTSC.

**Figure 3 ijerph-18-09509-f003:**
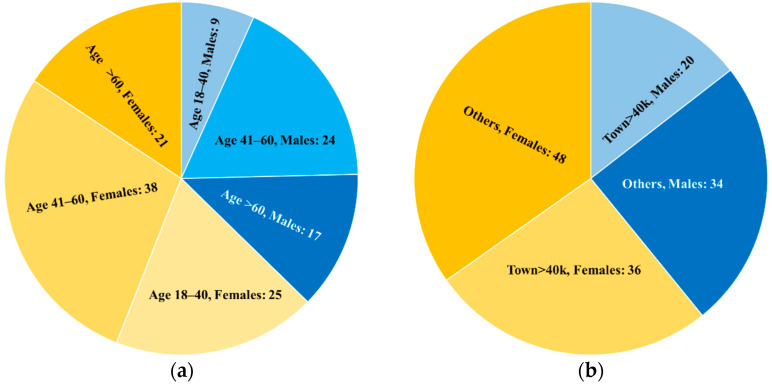
Basic demographic features of participants to forest therapy sessions: (**a**) Distribution by age group and gender; (**b**) Distribution by residence and gender.

**Table 1 ijerph-18-09509-t001:** Basic features of forest therapy sessions performed in Italy.

Session	Place, Municipality/Date/ Initial Time ^1^	Lat/ Lon/ Altitude	Weather, Temperature/ Tree Species/ Water Bodies—Streams	TVOC Average (Peak) Unit: μg/m^3^	Participants/ Valid Questionnaires ^2^/Gender ^3^
FTS1	Torre dei Sogni, Empoli and Montespertoli (FI)	43°41′11.77″ N	Sunny, weak wind, 30 °C	68 (120)	35
	22 August 2020	11°0′0.69″ E	Cypress, maritime pine, holm oak, lentisk		24
	14:30	100–200 m			M: 11; F: 13
FTS2	Monte Duro, Vezzano sul Crostolo (RE)	44°32′34.49″ N	Sunny, calm, 23 °C	27 (50)	35
	6 September 2020	10°32′26.49″ E	Scots pine, field maple, black and white hornbeam, elderberry, dogwood		21
	08:30	500–600 m			M: 8; F: 13
FTS3	Parco del Respiro, Fai della Paganella (TN)	46°10′33.25″ N	Sunny, calm, 22 °C	38 (108)	39
	11 September 2020	11°4′39.98″ E	Spruce, beech		33
	13:30	900–1000 m	Water streams, waterfalls		M: 8; F: 25
FTS4	Parco del Respiro, Fai della Paganella (TN)	46°10′33.25″ N	Sunny, calm, 17 °C	28 (65)	22
	12 September 2020	11°4′39.98″ E	Spruce, beech		17
	08:30	900–1000 m	Water streams, waterfalls		M: 5; F: 12
FTS5	Battisti Refuge, Ligonchio (RE)	44°15′42.24″ N	Cloudy, windy, 4 °C	10 (20)	9
	27 September 2020	10°24′45.58″ E	Beech, silver fir, mountain pine		8
	08:00	1750 m			M: 5; F: 3
FTS6	Torre dei Sogni, Empoli and Montespertoli (FI)	43°41′11.77″ N	Partly cloudy, weak wind, 15 °C	50 (90)	21
	17 October 2020	11°0′0.69″ E	Cypress, maritime pine, holm oak, lentisk		18
	14:00	100–200 m			M: 11; F: 13
FTS7	Fonte del Borbotto, San Godenzo (FI)	43°52′57.78″ N	Sunny to cloudy, calm, 7 °C	15 (30)	33
	18 October 2020	11°41′10.65″ E	Beech, scattered silver fir		29
	09:00	1250–1350 m	Water streams, pond		M: 12; F: 17
FTSC	Villa Vogel urban park, Firenze (FI)	43°46′33.54″ N	Sunny, calm, 28 °C	15 (56) ^4^	13
	23 June 2021	11°12′23.89″ E	Holm oak, stone pine and other species		12
	09:00	39 m	Water pond		M: 8; F: 4

^1^ Local solar time. ^2^ Number of self-assessment questionnaires further considered for analysis. ^3^ Data referred to valid questionnaires (M: male; F: female). ^4^ Urban park surrounded by buildings and trafficked roads. TVOC levels could not be considered representative of plant emissions.

**Table 2 ijerph-18-09509-t002:** Data about age, height and weight of participants to each forest therapy session.

Variable	Session	Number of Available Data	Mean ± Standard Deviation
Age	FTS1	24	52.42 ± 10.32
	FTS2	21	52.86 ± 9.89
	FTS3	33	45.58 ± 16.84
	FTS4	16	44.94 ± 15.15
	FTS5	8	64.38 ± 7.89
	FTS6	17	50.76 ± 14.37
	FTS7	29	51.83 ± 11.69
	FTSC	12	57.67 ± 15.74
Height	FTS1	23	170.74 ± 9.25
	FTS2	21	170.67 ± 9.03
	FTS3	33	165.36 ± 8.22
	FTS4	16	164.13 ± 7.45
	FTS5	8	166.25 ± 8.46
	FTS6	17	170.29 ± 11.54
	FTS7	29	170.28 ± 11.32
	FTSC	12	170.92 ± 8.84
Weight	FTS1	24	69.04 ± 10.59
	FTS2	21	70.52 ±13.24
	FTS3	33	58.97 ± 15.36
	FTS4	16	57.56 ± 6.46
	FTS5	8	71.13 ± 11.43
	FTS6	16	70.06 ± 13.98
	FTS7	29	69.62 ± 19.47
	FTSC	12	72.08 ± 16.75

**Table 3 ijerph-18-09509-t003:** POMS pre-intervention domain scores.

Domain	Session	Number of Available Data	Mean ± Standard Deviation	95% CI
Anxiety	FTS1	24	2.125 ± 3.687	(0.686; 3.564)
	FTS2	21	2.714 ± 2.918	(1.176; 4.252)
	FTS3	33	2.879 ± 2.955	(1.652; 4.106)
	FTS4	17	4.180 ± 4.680	(2.47; 5.89)
	FTS5	8	2.750 ± 2.605	(0.258; 5.242)
	FTS6	18	3.833 ± 3.682	(2.172; 5.495)
	FTS7	29	4.586 ± 4.280	(3.277; 5.895)
	FTSC	12	1.750 ± 2.261	(−0.285; 3.785)
Depression	FTS1	24	0.583 ± 1.886	(−0.515; 1.682)
	FTS2	21	1.000 ± 1.703	(−0.175; 2.175)
	FTS3	33	2.303 ± 3.127	(1.366; 3.240)
	FTS4	17	2.353 ± 3.807	(1.047; 3.658)
	FTS5	8	1.500 ± 2.268	(−0.403; 3.403)
	FTS6	18	2.500 ± 3.698	(1.231; 3.769)
	FTS7	29	2.069 ± 2.434	(1.069; 3.068)
	FTSC	12	0.750 ± 1.485	(−0.804; 2.304)
Anger	FTS1	24	0.958 ± 2.579	(0.138; 1.779)
	FTS2	21	1.000 ± 1.761	(0.123; 1.877)
	FTS3	33	1.455 ± 1.922	(0.755; 2.155)
	FTS4	17	1.529 ± 2.004	(0.554; 2.505)
	FTS5	8	1.250 ± 1.753	(−0.172; 2.672)
	FTS6	18	1.500 ± 2.407	(0.552; 2.448)
	FTS7	29	1.724 ± 1.869	(0.977; 2.471)
	FTSC	12	0.583 ± 1.443	(−0.577; 1.744)
Confusion	FTS1	24	1.792 ± 2.604	(0.670; 2.914)
	FTS2	21	1.905 ± 1.998	(0.705; 3.104)
	FTS3	33	2.939 ± 2.487	(1.982; 3896)
	FTS4	17	2.882 ± 3.257	(1.549; 4216)
	FTS5	8	2.000 ± 2.070	(0.057; 3.943)
	FTS6	18	3.889 ± 3.179	(2.593; 5.185)
	FTS7	29	3.310 ± 3.547	(2.290; 4.331)
	FTSC	12	1.500 ± 1.784	(−0.087; 3.087)

**Table 4 ijerph-18-09509-t004:** Statistical significance of outcomes from forest therapy sessions aggregated intervention groups (FTS1 to FTS7) and control group (FTSC).

Session	Anxiety	Depression	Anger	Confusion
Intervention groups (FTS1 to FTS7)	↓ F = 59.62; *p* < 0.001	↓ F = 24.64; *p* < 0.001	↓ F27.48; *p* < 0.001	↓ F = 30.98; *p* < 0.001
Control group (FTSC)	↔ F = 1.82; *p* = 0.191	↔ F = 0.74; *p* = 0.399	↔ F = 0.93; *p* = 0.345	↔ F = 0.43; *p* = 0.521

**Table 5 ijerph-18-09509-t005:** Results of the Ryan–Joiner test for normality of change-from-baseline for any mood state in any forest therapy session. NN = non-normal; all other distributions were normal.

Session	Anxiety	Depression	Anger	Confusion
FTS1	NN	NN	NN	
FTS2		NN		
FTS3	NN			
FTS4	NN			
FTS5				
FTS6		NN		
FTS7		NN		
FTSC				

**Table 6 ijerph-18-09509-t006:** Statistically significant differences between outcomes from each forest therapy sessions FTS1 to FTS7 and outcomes from the control session FTSC. Green arrows: *p* < 0.05. Blue arrows: 0.05 ≤ *p* < 0.10 (close to significance).

Session	Anxiety	Depression	Anger	Confusion
FTS1				
FTS2				
FTS3		↓ (*p* = 0.084)	↓ (*p* = 0.023)	↓ (*p* = 0.082)
FTS4			↓ (*p* = 0.028)	↓ (*p* = 0.093)
FTS5				
FTS6	↓ (*p* = 0.027)		↓ (*p* = 0.088)	↓ (*p* = 0.024)
FTS7	↓ (*p* = 0.042)		↓ (*p* = 0.072)	

**Table 7 ijerph-18-09509-t007:** Factors and respective values used in the mixed model ANOVA for the assessment of the association of forest therapy outcomes (sessions FTS1 to FTS7) with sociodemographic traits.

Factor	Values
Age	From 18 onwards (integer)
Height	Any (integer)
Weight	Any (integer)
Job	IN; MA; RE; SO ^1^
Education	SS; HS; UN ^2^
Residence	CI; VI ^3^
Smoke	0; 1 ^4^
Sport ^5^	0; 1; 2; 3 ^6^

^1^ IN: intellectual worker; MA: manual worker; RE: retired; SO: social worker. ^2^ SS: secondary school; HS: high school; UN: university and higher. ^3^ CI: city greater than 40,000 inhabitants; VI: village below 40,000 inhabitants. ^4^ 0: not smoker; 1: smoker. ^5^ Sport is understood as a session of physical activity, indoor or outdoor, regardless of the duration and intensity. ^6^ 0: no sport; 1: 1 to 2 times a week; 2; 3 to 4 times a week; 3: more than 4 times a week.

## Data Availability

All data are available by contacting the corresponding authors.

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
