# Peer review of "Short-Term Effects of Forest Therapy on Mood States: A Pilot Study"

_ijerph, 2021, doi:10.3390/ijerph18189509_

Round 1
Reviewer 1 Report
I have read the authors’ reply to my comments and I find that they give satisfying answers to all my points. I accept them.
I think the authors did their best to address all the critics raised by the reviewers.
Each limiting factor that could not be changed or augmented has been mentioned as limitation of the experimental study, which is recategorized and renamed as a pilot-study, as it is indeed.
An extra effort has also been made in creating a suitable control group and experiment ie. the forest bathing in an urban park (though the number of participants is a bit low, but this is also listed as a limitation). The whole article is restructured, additional, previously lacking, information is supplied, and the statistics is completely reperformed.
This way, new perspectives have also emerged as to aggregate data of the seven remote FTS-s, compare them to the FTSC, furthermore to analyse specific characteristics of the single FTS-s, which may cause the alternating significancies in mood states.
I especially liked the introduction of the concept of environmental coherence, for I believe the effects of forest bathing/therapy must be multifactorial.
I have one tiny remark on Table 7 regarding sport: ”4…2: 2 to 4 times a week” is I guess 3 to 4 times a week; and please also specify your definition for “sport” (duration and intensity). What does doing sport x times a week mean?
English language is excellent.
Altogether, now in this revised form, in my opinion the manuscript (and the work itself) has been significantly improved and warrants publication in the IJERPH.
Author Response
Response to Reviewer 1 Comments
I have read the authors’ reply to my comments and I find that they give satisfying answers to all my points. I accept them.
I think the authors did their best to address all the critics raised by the reviewers.
Each limiting factor that could not be changed or augmented has been mentioned as limitation of the experimental study, which is recategorized and renamed as a pilot-study, as it is indeed.
An extra effort has also been made in creating a suitable control group and experiment ie. the forest bathing in an urban park (though the number of participants is a bit low, but this is also listed as a limitation). The whole article is restructured, additional, previously lacking, information is supplied, and the statistics is completely reperformed.
This way, new perspectives have also emerged as to aggregate data of the seven remote FTS-s, compare them to the FTSC, furthermore to analyse specific characteristics of the single FTS-s, which may cause the alternating significancies in mood states.
I especially liked the introduction of the concept of environmental coherence, for I believe the effects of forest bathing/therapy must be multifactorial.
English language is excellent.
Altogether, now in this revised form, in my opinion the manuscript (and the work itself) has been significantly improved and warrants publication in the IJERPH.
Response: The authors are sincerely grateful to the esteemed Reviewer for her/his outstanding job, which contributed substantially to improving the manuscript to a level that is considered publishable in this important Journal.
Point 1: I have one tiny remark on Table 7 regarding sport: ”4…2: 2 to 4 times a week” is I guess 3 to 4 times a week; and please also specify your definition for “sport” (duration and intensity). What does doing sport x times a week mean?.
Response 1: The esteemed Reviewer is right about the frequency, which has been corrected: “2: 3 to 4 times a week”. Moreover, the definition for “sport” is introduced in a footnote to Table 7: “5 Sport is understood as a session of physical activity, indoor or outdoor, regardless of the duration and intensity”.
In the Discussion section, it is pointed out that “Significant association with absence of sport could mitigate missing information on duration and intensity, according to the definition of sport presented in Table 7” (lines 577-579). However, such missing information represents another limitation, which is declared again in the Discussion section: “Moreover, further information on the duration and intensity of sport, understood as a session of physical activity, would be useful to refine the analysis” (lines 664-665). The authors gratefully acknowledge the esteemed Reviewer for pointing out this issue.
Reviewer 2 Report
Dear Authors,
Despite the fact that you did not care for a control group and did not conduct an additional study to have it, you did a very good revision and I believe that the manuscript may be published in present form.
Congratulations and good luck with further research (remember at least one well-thought-out control group in them in advance).
Best,
Author Response
Dear Authors,
Despite the fact that you did not care for a control group and did not conduct an additional study to have it, you did a very good revision and I believe that the manuscript may be published in present form.
Congratulations and good luck with further research (remember at least one well-thought-out control group in them in advance).
Response: The authors are sincerely grateful to the esteemed Reviewer for her/his outstanding job, which contributed substantially to improving the manuscript to a level that is considered publishable in this important Journal. However, additional experimental research was performed with a control group (control session called FTSC), and it was presented and discussed in the manuscript, including the limitation due to the relatively scarce number of participants. Further research will involve more control groups with higher number of participants.
This manuscript is a resubmission of an earlier submission. The following is a list of the peer review reports and author responses from that submission.
Round 1
Reviewer 1 Report
The manuscript assigned for me to peer-review is a original research manuscript titled as "ijerph-1247157: Short-term Psychological Effects of Forest Therapy: Early Evidence in Italy".
The authors of this manuscript have described interesting study results on preliminary evidence of forest immersion arose different functionality towards specific psychological indexes conditioned at least on gender and age groups, as well as meteorological comfort, structured programs and, possibly, volatile organic compounds. The authors tried to show and produce beneficial effects of forest environment immersion (forest therapy) to human health, in particular psychophysical relaxation. I think their approach will suggest several hints and useful approach to the relevant researches and public health policies.
The authors could make major revisions as followings:
1) PAGE 2 LINE 48: The sentence "... , and spiritual wellbeing." seemed to be omitted its references (maybe, ref. 1-4 ?).
2) PAGE 2 LINE 61: The sentence "... mental and physiological health." seemed to be omitted its references (maybe, ref. 8?).
3) PAGE 2 LINE 68: The sentence "... the season and time of the day." seemed to be omitted its references.
4) PAGE 2 LINE 70: The sentence "... extent of the effects." needs to be more clealy described (eg. "extent of beneficial effects of the forest immersion").
5) PAGE 2 LINE 77: The sentence "... throughout the year." seemed to be omitted its references.
6) PAGE 2 LINE 98: The sentence "... to protocols aimed at boosting the health outcomes" seemed to be omitted its references.
7) PAGE 2 LINE 138: The sentence "... and gains in terms of productivity." seemed to be omitted its references.
8) PAGE 2 LINE 140: In the mid of sentence, "participants ," contained unecessary space.
9) PAGE 3 LINE 152: In the "2. Materials and Methods" section, although concise subjects demographics shown in the Table 1 and Figure 2, authors should describe the enrolled subjects characteristics and related demographic data and relevant enrollment process with informed consents, firstly.
10) PAGE 8 LINE 303: Authors should take statistical expert consultations with conventional statistical packages for precise data analyses. Although MS Excel provides powerful graphical and statistical support function, it is not a proper statistical package for precise statistical analyses. Freely available, open source R-statistics packages, or comercial SPSS can be used, maybe. If the authors must use the Excel, they should clarify and describe the formula they applied with proper references, and version number and product information according to the accademic citation rule. Furthermore, the subjects group size of this study is small (most of the valid questionnaires n<30, according to Table 1), so the authors should check the normality of the subject distributions, and should select non-parametrical analysis methods. Because this study did not enrolled the control group, Student's t-test should not be used and the authors should be perform before-after forest therapy effect comparisons according to the data distribution normality status using "paired Student t-test" or "Wilcoxon signed-rank test".
10) In the Results section, the authors need to descrbe and show the results of analyses on effects of altitude, wheather, temperature, and TVOCs.
11) In the Discussion section, the authors may refer and discuss the health effects of forest therapy on psychophysical relaxation with more clinical study results, such as sleep quality (eg. "Kim HY, et al. An Exploratory Study on the Effects of Forest Therapy on Sleep Quality in Patients with Gastrointestinal Tract Cancers. Int J Environ Res. Public Health 2019, 16(14), 2449; https://doi.org/10.3390/ijerph16142449"). And more detailed description for limiation of this study.
12) PAGE 18 LINE 727: In the Conclusion section, the paragraph including several citations, "The mobilization of financial resources ... useful model." should be moved to the Discussion section.
13) Too many references should be reduced in consideration of original research article form.
Good luck!
Reviewer 2 Report
This manuscript addresses the interesting issue of the importance of forest therapy and may interest the readers. However, it cannot be published in this form - it requires significant changes.
Major remarks:
- The introduction ends with bulleted "The main motivations of this study". This is not a good strategy. Describe your goals and formulate specific hypotheses. If your main goal was "Preliminary functional qualification of sites for forest therapy practices in Italy" please publish this manuscript in your local Italian journal. Or give important theoretical problems and hypotheses.
- The participants are not described extensively. You give only rudimentary data. Describe it carefully, e.g., recruitment details, inclusion and exclusion criteria.
- I don't understand the strategy of statistical analysis. Why didn't you use ANOVA? Why the t-test? Why use difference scores in major analysis rather than in post-hoc? Why are you not reporting test results and only providing effect sizes?
- The control group problem is significant here and your justification is not convincing. The control group should experience a very different forest therapy, as well as performing tasks of similar physical intensity in the forest, without elements of forest therapy. I propose to carry out such control tests.
- The work lacks references to a huge European project on the impact of nature on cognitive processes: Fostering knowledge about the relationship between Information and Communication Technologies (ICT) and Public Spaces. I am thinking in particular of the book on learning in cyberparks and the work in the field of cognitive neuroscience relating to the impact of activities close to nature on the functioning of the brain.
Minor remarks:
- Provide exact details of the consent of the ethics committee (number, name, etc.)
- What software was used to analyze the data?
- There are too many bulleted lists in this manuscript. Objectives, results, conclusions are presented in this way. Describe them carefully, do not use bulleted lists - it's not a textbook for students.
- All the figures are of poor quality, their fonts are too small, and the contrasts are wrong.
Reviewer 3 Report
Short-term Psychological Effects of Forest Therapy: Early Evidence in Italy
I was happy to read this thorough, reasonable, and valuable work that enriches the European literature on forest therapy. The authors did make great effort to contribute to the field, and I liked the concept of the article to compare various factors of the different forest environments (and, also some of the participants) that may influence the outcome, that is the effect on people. It really is desirable to connect medical and forest research on behalf of both humans and the natural environment.
I have the following comments and suggestions for minor revision:
1) When reading only the Abstract, despite the title (short-term) it was not obvious to me whether the seven forest therapy sessions included the same participants under different conditions, or different individuals in independent sessions. Please make it clear that it is the latter case.
2) A) The gender, age (and residence) of the subjects and that they are “admittedly healthy” is somehow scant. Unfortunately, no physiological parameters were measured, whose lack may verify the missed exclusion of major diseases and checking for basic health parameters, but at least the exclusion of acute illnesses as well as of alcohol and drug abuse and psychological or psychiatric treatment should have been done and written down in the text.
- B) The fact that “participants were allowed to attend only one session, to prevent any contribution” is a very well criterion. Since the Italian Alpine Club was also among the organizers/authors, were its members also not allowed to take part in the sessions, for the same reason? Were the regular forest visitors excluded for a possible persisting effect?
- C) Since all sessions were performed in remote forest areas (on the one hand this is good to study the pure forest effects, on the other hand it is not so realistic in a way that it is hard to incorporate into everyday life or lifestyle, if later suggested, also it is hard to recruit participants), so how was the transportation carried out? Was it not stressful for the subjects to approach the sites?
3) One of the major findings of the study is the great contribution to the beneficial outcomes of the professional guidance by a psychologist or psychotherapist, since the most effective FTS2-3-4-6-7 sessions were all guided, while the single non-guided session, FTS1, as a control, produced insignificant results. This single control session seems a bit insufficient to me. When comparing FTS1 and FTS6 (lines 516-523), the same site and high levels of TVOCs are all right, but I would not call both 30°C (FTS1) and 15°C (FTS6) moderate meteorological comfort, since there is 100% difference between them. 30°C may make people impatient and tired; the difference in temperature may also have contributed to the altering results, please mention this.
4) Finally, when comparing to the healthcare stuff study [38] (from line 446) and, also in general, I think the remoteness, the naturalness, and the fascinating site in the current study may also play a role in the greater significance of the results compared to literature data.